# Hip Joint Angles and Moments during Stair Ascent Using Neural Networks and Wearable Sensors

**DOI:** 10.3390/bioengineering10070784

**Published:** 2023-06-30

**Authors:** Megan V. McCabe, Douglas W. Van Citters, Ryan M. Chapman

**Affiliations:** 1Thayer School of Engineering, Dartmouth College, Hanover, NH 03755, USAdouglas.w.van.citters@dartmouth.edu (D.W.V.C.); 2Department of Kinesiology, University of Rhode Island, Kingston, RI 02881, USA

**Keywords:** hip joint, wearable sensor, wearables, machine learning, neural networks, inertial measurement units, instrumented insoles, OpenSim, biomechanics

## Abstract

End-stage hip joint osteoarthritis treatment, known as total hip arthroplasty (THA), improves satisfaction, life quality, and activities of daily living (ADL) function. Postoperatively, evaluating how patients move (i.e., their kinematics/kinetics) during ADL often requires visits to clinics or specialized biomechanics laboratories. Prior work in our lab and others have leveraged wearables and machine learning approaches such as artificial neural networks (ANNs) to quantify hip angles/moments during simple ADL such as walking. Although level-ground ambulation is necessary for patient satisfaction and post-THA function, other tasks such as stair ascent may be more critical for improvement. This study utilized wearable sensors/ANNs to quantify sagittal/frontal plane angles and moments of the hip joint during stair ascent from 17 healthy subjects. Shin/thigh-mounted inertial measurement units and force insole data were inputted to an ANN (2 hidden layers, 10 total nodes). These results were compared to gold-standard optical motion capture and force-measuring insoles. The wearable-ANN approach performed well, achieving rRMSE = 17.7% and R^2^ = 0.77 (sagittal angle/moment: rRMSE = 17.7 ± 1.2%/14.1 ± 0.80%, R^2^ = 0.80 ± 0.02/0.77 ± 0.02; frontal angle/moment: rRMSE = 26.4 ± 1.4%/12.7 ± 1.1%, R^2^ = 0.59 ± 0.02/0.93 ± 0.01). While we only evaluated healthy subjects herein, this approach is simple and human-centered and could provide portable technology for quantifying patient hip biomechanics in future investigations.

## 1. Introduction

Quantifying hip kinematics and/or kinetics is critical for evaluating pathologies of and treatments for the hip joint. One of the most common pathology/treatment pairs is symptomatic hip osteoarthritis (OA) [1], and the end-state treatment for this disease, total hip arthroplasty (THA) [2,3,4]. THA is incredibly common, with nearly 500,000 annual THA procedures in the United States [5,6]. Characterizing hip kinematics/kinetics during common activities of daily living (ADL), e.g., standing from a chair, walking, stair ascent, [2,3,4,7] allows for more realistic in vitro and in silico simulations of hip OA, which can improve diagnostic and treatment methods (e.g., implant design and gait rehabilitation) [3,8,9,10,11,12,13,14,15,16].

Current gold-standard methods for quantifying healthy and pathological hip angles/moments are mostly limited to expensive and non-portable laboratory/clinic-based data captures. Typically, biomechanical modeling software (e.g., OpenSim) is used to compute joint kinematics/kinetics using captured force plate and retroreflective or active marker optical motion capture (MOCAP) data [17,18]. However, this approach is limited by its high cost (e.g., force plates > USD 10K [19] and MOCAP fixed cost > USD 14K [20,21]) and non-portability.

Contrastingly, wearable sensors and force-instrumented insoles offer significantly lower-cost portable modalities for monitoring biomechanics data in real-world settings outside of the clinic/laboratory [22,23,24]. Inertial measurement units (IMUs) provide a portable means for quantifying joint kinematics using small, electromechanical sensors that are affixable to body segments. Force-measuring insoles are worn inside shoes to quantify vertical GRFs during repeated gait cycles. Both sensing modalities facilitate high-fidelity capture of kinematics/kinetics both inside and outside of laboratory/clinical settings. In traditional biomechanical modeling, this requires precise fixation to patients, is time consuming, and necessitates detailed coordinate transformations to transform wearable data captured in local coordinate frames to an anatomic frame of reference for kinematics/kinetics [25,26,27]. However, prior work in our own lab leveraged a machine learning approach to calculate 2D (sagittal, frontal planes) joint angles/moments from one shin-mounted IMU, one force-instrumented insole, and a simplified artificial neural network (ANN) with two hidden layers and five nodes per layer [28].

While highly successful (relative root mean square error (rRMSE) = 15%; relative to predicted average and ground truth moment ranges; R^2^ = 0.85), upright walking was the only activity captured in that investigation. Although that task is common and easy to capture, it may not represent the most critical ADL to monitor before or after THA. In contrast, stair ascent requires increased function (i.e., greater range of motion (ROM), improved balance, etc.) before/after surgery and may represent a more specific and sensitive ADL for evaluating function in these patients. Accordingly, the focus of this study utilizes a similar approach to create and validate a ML algorithm (wearable-ANN approach) capable of computing 2D (sagittal/frontal plane) hip joint kinematics/kinetics using IMUs affixed to the shin and thigh and one force-instrumented insole under the ipsilateral foot during stair ascent. As in other studies that develop computational algorithms, our investigation used healthy individuals as a first attempt in creating this novel approach [29,30,31,32]. All three sensors (2 × IMUs, 1 × force-instrumented insole) had input data streams for an ANN (2 hidden layers, 10 total nodes) (Figure 1 Bottom). The performance of this algorithm was compared with gold-standard optical MOCAP and force-measuring insole data inputs for OpenSim (Figure 1 Top). Force-instrumented insoles were selected in favor of force plates in order to collect repeated gait cycles while performing stair ascent. As in prior investigations, the wearable-ANN approach would be considered successful if average rRMSE was less than 13% [28,31]. Thus, we hypothesized that all variables (sagittal angle, sagittal moment, frontal angle, frontal moment) would achieve rRMSE < 13%.

## 2. Materials and Methods

### 2.1. Data Capture

At the highest level, data collection involved quantifying subjects’ height/weight, attaching and performing calibrations as recommended by the manufacturers for all sensing modalities (Figure 2), and recording data during subject performance of stair ascent. In total, 17 healthy subjects (10 M; 26.8 ± 6.4 years; 1.74 ± 0.08 m; 81.6 ± 19.5 kg) were recruited, consented, and enrolled from the local university population after institutional review board approval (inclusion criteria: age ≥ 18 years, no musculoskeletal/neuromuscular impairments impacting lower extremities, no terminal illness resulting in death within one year, clinical full hip extension ≥ 10° and flexion ≥ 100°, and complete participation in the study) [33].

More specifically, three sensing modalities were used (optical MOCAP, IMUs, and force-instrumented insoles). Subjects then had retroreflective optical MOCAP markers adhered to their skin over bony landmarks according to the modified lower body Helen Hayes markers (Figure 2A) [34,35]. Six S250e cameras (OptiTrack Motive Body 1.10, NaturalPoint, Inc., Corvallis, OR, USA) were calibrated according to the manufacturer’s instructions. Three-dimensional optical MOCAP marker data were temporally synchronized during all trials and stored on the local computer for processing after data capture. Subjects then had two IMUs (Figure 2B; APDM v1 Emeralds, APDM Inc.; Portland, OR, USA; f_s_ = 128 Hz) attached to their dominant anteromedial shank and lateral thigh via silicone backed Velcro straps (Waterloo Footedness Questionnaire, 13 right-footed) [36]. IMUs were calibrated per manufacturer’s instructions and logged inertial data continuously throughout all stair ascent trials. Lastly, subjects were fitted with previously validated force-instrumented insoles, called “Loadsols”, to measure vertical GRFs (Figure 2C; Novel Electronics, St. Paul, MN, USA; f_s_ = 100Hz). Loadsols wirelessly streamed vertical GRF data over Bluetooth to our laboratory iPad (Apple, Cupertino, CA, USA) for offline post-processing.

After subjects were fitted with all three sensor types (optical MOCAP, IMU, force insoles), they completed 10 s stationary standing used for OpenSim model scaling and 3 × 10 s stair ascent trials at a set moderate pace (Speed 8, StairMaster StepMill 7000PT, Vancouver, WA, USA; 20.32 cm rise × 23.5 cm run). Stair ascent was captured on a stair exercise machine to facilitate capture of repeated, successive gait cycles in laboratory (average gait cycles/subject/trial: 4.53 ± 0.62).

### 2.2. Data Pre-Processing

#### 2.2.1. Overview

Data were pre-processed to prepare raw data for respective workflows (OpenSim traditional biomechanical modeling workflow with optical MOCAP and insole data versus MATLAB ANN development/validation workflow with IMU and insole data (see Figure 1)). For all subjects, the stair ascent trial with the least MOCAP marker trajectory loss was utilized for final analysis. MATLAB scripts were written for filtration and temporal synchronization using gait-cycle analysis (Figure 3) across all sensing modalities. Finally, filtered and synchronized data were separated and passed to the respective workflows (traditional modeling vs. ANN development/validation).

#### 2.2.2. MATLAB Pre-Processing

Optical MOCAP marker trajectories were edited in Optitrack Motive software interpolating/filling trajectory gaps and filtering data (low-pass Butterworth, f_cutoff_ = 6 Hz) prior to importing into MATLAB. An additional filter was created in MATLAB for Loadsol and IMU data (low-pass Butterworth, f_cutoff_ = 10 Hz). In MATLAB, Loadsol data were then resampled (100→128 Hz) [23,30] to normalize the capture frequency across sensing modalities. IMU data trials were separated using a manufacturer-supplied button connected to a third IMU controlled by the research team during data collection to create trial “start” and “end” timepoints. Utilizing a process deployed by Coley et al. and Yang et al., shank IMU data used for gait analysis were filtered at 2.3 Hz rather than 10 Hz and were subsequently rotated using a rotation matrix from the goniometrically captured rotation angle from real IMU position to pure lateral position (121.3 ± 6.6°) [37,38].

#### 2.2.3. Temporal Synchronization (Figure 3)

Synchronization with respect to time was achieved across sensor types using previously published gait-based synchronization techniques. First, the initial heel strike (T_HS_) was located for all sensing modalities. All sensing modalities were then temporally shifted to align that time point with MOCAP T_HS_. All files were then truncated for consistent data length between sensors. For MOCAP, T_HS_ was identified by locating the point at which anteroposterior velocity of the heel marker = 0 m/s immediately followed the swing phase (Figure 3, top) [39]. To compute HS for Loadsols, we utilized a well-established threshold of 20 N [39,40,41]. Accordingly, for Loadsols, T_HS_ occurred when data exceeded the 20 N threshold with a positive slope (Figure 3, middle). Toe off was located similarly when the Loadsol data descended to less than 20 N. Loadsol heel strikes and toe offs defined each gait cycle’s percentages (stance % = 67.0 ± 3.4%). T_HS_ for IMUs was located via the first of two local minima of the shank’s mediolateral axis angular velocity during the stance phase (Figure 3, bottom) [37,38,42].

#### 2.2.4. OpenSim Workflow

Calculating hip angles/moments for the gold-standard method was completed via OpenSim with MOCAP and Loadsol data as inputs [17,18]. Scaling, IK, and ID settings were selected via use of OpenSim Tutorial 3 [43]. First, standing trial data were processed and scaled, fitting the generic gait2392 model to the specific anatomy of each subject [44]. The height of each subject-specific model was manually scaled given that no MOCAP markers were utilized on the upper body [45]. The scaled model and MOCAP data from stair ascent trials were combined together to compute the joint angles [46]. Vertical GRFs from Loadsol were then applied to the talus defined by the midpoint between the MOCAP ankle markers [47]. These data were then combined with IK joint angles in the ID tool to compute the net joint moments. Finally, IK (joint angles) and ID (joint moments) results were imported into MATLAB for gait-cycle normalization (stride time to gait percent). Data were averaged across the cohort (ensemble averaged) and across gait cycles using the gait percentage vector.

#### 2.2.5. ANN Workflow

A shallow (2 hidden layers) feed forward ANN was created via a subject-general approach for the wearable-ANN workflow. Leave-one-out cross-validation (LOO-CV) trained the ANN by withholding one subject’s data for validation. Training rounds involved 17 repetitions, allowing all subjects to function as the validation dataset with inputs, including activity duration, dominant foot vertical GRF, and IMU data. Each IMU contained 13 inputs (3D acceleration, angular velocity, and magnetic field; 4D orientation quaternion) resulting in 3840 individual data points per IMU per trial (30 s trial, 128 Hz). The output layer of the ANN developed in MATLAB computed sagittal/frontal plane hip angles and moments. These were subsequently stored as a series of arrays of the respective metric during each gait cycle.

Prior work was utilized as a model for ANN architectures herein [30,31,32]. Hidden layers were created via hyperbolic tangent functions, whereas the output layer was created by a linear function. Hyperbolic tangent functions are more sensitive to input data stream variability and afford a greater working range than commonly used sigmoidals [30,32,48,49]. Although larger ANNs can model complex relationships more effectively, they are prone to longer computations and overfitting training data. Previous studies (<20 subjects, <150 nodes) [30,32] have facilitated optimizing the ANN to an architecture with two hidden layers and five nodes per layer, using all previously described inputs.

LOO-CV was implemented during training by first normalizing all ANN inputs (range = −1, 1) with the map-minmax function in MATLAB. Then, the ANN was initialized via the Nguyen and Widrow function (initnw in MATLAB). Finally, the ANN was trained using the Levenberg–Marquardt algorithm (trainlm in MATLAB). Repeated training allowed each subject to perform as the validation set. Performance metrics were averaged across separately trained/tested ANNs (17 subjects, 10 training rounds = 170 trained and tested ANN iterates) [30,31,32]. Initnw filled the weights/biases with initial values (“initialization”) and trainlm backpropagated/optimized until the gradient stopped decreasing for >6 training passes.

To evaluate ANN performance, coefficients of determination (R^2^) and rRMSE were calculated by comparing the gold-standard workflow to the wearable-ANN workflow. ANN performance was averaged across 170 training iterates. Subject-specific ANN performance was evaluated via violin plots. The shape of the violin was determined via the kernel density estimation of the probability distribution curve of the dataset with statistical outlier removal. Paired *t* tests were then conducted on the average angles and moments between the two workflows (gold standard vs. wearable-ANN) at critical percents of the gait cycle, including heel strike (0%), mid-stance (30%), toe off (~60%), and mid-swing (80%).

## 3. Results

### 3.1. Stair Ascent on the Exercise Machine

Sagittally, stair ascent required greater hip ROM than typically required during walking (55.6°) but lower peak hip flexion moments (1.0 Nm/kg). In the frontal plane, stair ascent exhibited greater ROM (17.4°) and peak abduction moments (1.6 Nm/kg) than typically seen during walking. The first of two abduction peaks during stair ascent stance was slightly larger than the second.

### 3.2. Gold-Standard Approach

Average hip angles (Figure 4A, top; Table 1) and moments (Figure 4A, bottom; Table 1) of the group are displayed with ±1 SD (shaded) throughout the stair ascent gait cycle. The gold-standard angles/moments had similar shape to previously published literature values but with larger peak moments [50,51,52]. Like walking, peak extension/flexion moments occurred at ~20%/~50% of the gait cycle, respectively [28]. In the frontal plane, the gold-standard approach resulted in the expected double abduction peak during stance.

### 3.3. Wearable-ANN Approach

Thick curves in Figure 4A demonstrated that gold-standard and wearable-ANN computed angles/moments both showed consistent curvature. Wearable-ANN peaks were consistently reduced relative to the gold-standard approach. Excluding adduction angle, rRMSE was <20% (Table 1). Adduction moment was most successfully predicted (R^2^ = 0.93, rRMSE = 12.7%). Adduction angle was most challenging to predict (R^2^ = 0.59, rRMSE = 26.4%). Statistically significant differences existed between the two computation methods for flexion angle and adduction moment at 0% of the gait cycle as well as for flexion moment at 80% of the gait cycle. While statistically significant, it is unclear if these differences are clinically relevant.

Violin plots (Figure 5) show wearable-ANN performance variability across subjects. Dots represent average subject performance as the validation set across rounds (white dot = median, grey bars = IQR). The wearable-ANN approach predicted sagittal plane angle well (i.e., wide, short violin near R^2^ = 0.9; R^2^ min~0.5); however, it had more variability predicting sagittal moment (i.e., elongated violin with two peaks at R^2^~0.9 and 0.6). Adduction moments were predicted with high accuracy (0.8 < R^2^ < 0.99). Predicting adduction angles varied more widely across subjects (0 < R^2^ < 0.92).

## 4. Discussion

### 4.1. Overview

Our overarching objective herein was to develop and validate a machine-learning-based method for quantifying 2D hip joint kinematics and kinetics in healthy individuals during stair ascent. We hypothesized that all metrics (sagittal angle, sagittal moment, frontal angle, frontal moment) would achieve rRMSE < 13%. However, our hypothesis was rejected for three of the four metrics (sagittal angle = 17.7%, sagittal moment = 26.4%, frontal angle = 14.1%). The remaining metric (frontal moment) hypothesis failed to be rejected (rRMSE = 12.7%). Although our hypotheses were rejected in three measures, much of the increase in rRMSE was driven by differences between the methods that are likely not clinically significant (e.g., 0.1 Nm/kg for sagittal moment at 80% gait cycle). One notable statistically significant difference that may be clinically significant was the difference between the two computation methods for sagittal angle at 0% of the gait cycle (58.0° vs. 50.5°).

### 4.2. Stair Ascent on Exercise Machine

The stair exercise machine allowed for easy, continuous capture of repeated gait cycles. However, the steps were taller and shallower than standard steps. Moreover, they moved posteriorly as subjects ascended in contrast to standard stationary stairs. Previous work has shown that higher staircase inclines lead to increased hip flexion [53]. It is also likely that moving steps assist in hip extension and hamper hip flexion. Specifically, because the step is moving posteriorly, subjects do not need to exert as much force during hip extension but need to generate a large flexion moment to lift their leg up and to clear the next step. Further, the CoP error (discussed below) confounds robust characterization of the biomechanics of stair ascent on the exercise machine. Future work should model CoP movement on the steps, which would need to consider subjects with long feet “tip-toeing” due to shallow steps.

### 4.3. Gold-Standard Approach

The gold-standard approach for computing hip moments matched the curvature of previously conducted studies [2,50,52,53,54,55]. However, peak flexion moments were >25% larger than those calculated by Costigan et al. [50]. Most studies report no isolated peak flexion moment during stair ascent, but the gold-standard approach computed a substantial peak (1.0 ± 0.26 Nm/kg). These deviations may be explained by always placing the center of pressure where the talus is located. This methodological decision increases the vertical GRF lever arm during the initial/terminal stance by several centimeters (Figure 6). Previous work has characterized moment sensitivity to small errors in CoP, finding that 1 cm anterior shifts increase maximum extension moments by 8%, 1 cm posterior shifts increase maximum flexion moments by 16%, and 3 cm lateral shifts increase the double-peak abduction moments by 20% [56,57]. As such, improving how the CoP is modeled in this approach could improve the final moment results. Future work using our gold-standard approach should model CoP movement and validate the approach against optical MOCAP and force plates. One other potential improvement for this would be leveraging a similar approach to Chiu et al. Their work recommended quantifying the CoP path throughout the gait cycle as a percentage of that gait cycle as well as foot length percentage. This could be leveraged to estimate CoP without the need for direct measurement [58]. Additional approaches recommend subtracting a constant offset or leveraging wearable instrumented-insoles to compute CoP [59,60,61,62,63]. Perhaps pressure-sensing insoles are the most promising approach because developed algorithms can compute nonvertical GRF and CoP, which may be important for studying gait pathologies [64,65]. In general, the gold-standard method utilized herein matched well with prior work and allowed for comparison to the proposed wearable-ANN approach.

### 4.4. Wearable-ANN Approach

The proposed approach using wearable sensors and ANNs (2 hidden layers, 10 total nodes) achieved average R^2^ = 0.77 and rRMSE = 17.8% across all outputs and subjects. This was slightly higher than our original aim of the error metrics described by Mundt et al. of rRMSE < 13% [31]. The difference between the two methods is likely attributable to a number of factors, including some anomalous individual subjects (e.g., one subject rRMSE < 25%), portions of the gait cycle that have greater soft tissue noise (e.g., heel strike at 0% gait cycle), and differences that are statistically significant but that may not be clinically relevant (e.g., sagittal moment at 80% gait cycle). Although these differences are real, this ~5% difference in rRMSE values is likely not clinically relevant. Likely more critical and clinically valuable, however, may be angles/moments described at different time points during the stair ascent gait cycle (e.g., heel strike).

In an ideal setting, the ANN design for quantifying joint angles and/or moments would be leveraged biomechanical modeling. However, connecting biomechanical modeling (i.e., physics-based with machine learning approaches (i.e., ANN-based) has historically been difficult. This is especially true for subject-general approaches. Lim and her co-authors created a model with linked segments to guide creation of their ANN but eventually utilized trial and error to determine the appropriate number of hidden nodes [30]. In our opinion, the initial development of wearable-ANN approaches should likely instead acquire training data that more appropriately capture the variability expected in future experimental datasets. The results herein, and those presented by the previously noted Lim et al., indicate that shallow ANNs may be trained with relatively small datasets (<600 gait cycles) and still achieve success (average R^2^ = 0.77, rRMSE = 17.8% presented here).

Poor ANN performance indicates that the training data did not represent patterns in the validation set. The poorest prediction metric (adduction angle) seems to have greater variability across subjects than the other output metrics. The standard deviation of gold-standard computed adduction angle was larger than most other metrics, especially as a percent of the mean. The ANN approach yielded reduced standard deviation, suggesting that the ANN likely struggled with highly variable frontal kinematics. In contrast, the ANN method computed frontal plane moments most successfully (R^2^ > 0.90) and reported a standard deviation similar to that of the gold-standard computation method. Improving ANN performance may be accomplished by leveraging data augmentation to add variability into training data, as shown by Mundt et al. [31]. Another opportunity for improving this ANN approach could be found in customizing ANNs for specific subject categories (e.g., age range, gender, pathology) [66]. Finally, improving ANN performance could be accomplished via sensor types/quantities/placement optimization for particular applications.

### 4.5. Limitations

We leveraged a previously published method to quantify a movement beyond level ground walking. While this represents an exciting step forward in remote sensing capabilities, there are limitations. Notably, the error was greater than hypothesized across all subjects (rRMSE = 17.7% vs. 13%). Additionally, we only evaluated one movement in a well-controlled setting. Another limitation includes the use of a StairMaster compared to traditional stairs. Specifically, although we did not experience marker occlusion or other issues while tracking optical MOCAP markers, care must be taken when arranging cameras around a StairMaster. This may or may not be the same when capturing biomechanics data on traditional stairs. Finally, while these preliminary results are encouraging, we elected to conduct a statistical evaluation on discrete points in the gait cycle. The statistical surety of our findings could have been different if we conducted a statistical evaluation on the continuous data stream (e.g., SPM1D). These limits represent an additional opportunity for future work.

## 5. Conclusions

Despite the noted limitations of the present work, we have shown that the previously developed method for using wearable motion sensors and ANN to evaluate joint kinematics/kinetics is feasible in movements that are more complex than level-ground ambulation. This implies that this approach can be utilized to quantify a variety of ADL outside of well-controlled laboratory/clinical settings. When extrapolated beyond healthy subjects into patient populations, this has the potential to impact a variety of patient populations, including individuals receiving joint replacements. As telemedicine continues to advance, this has promise to become a primary modality of diagnosis and treatment development.

## Figures and Tables

**Figure 1 bioengineering-10-00784-f001:**
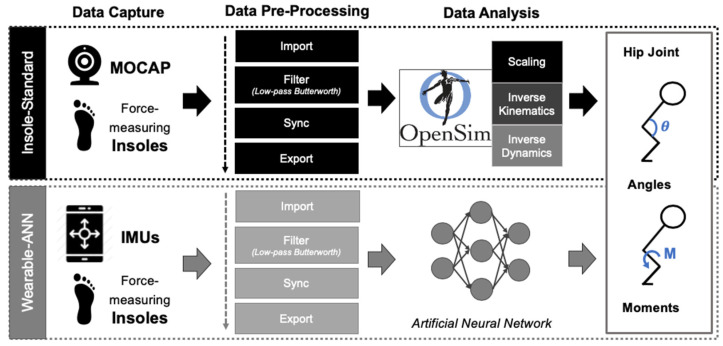
Overview of study workflow. Three raw datasets were captured: (1) force-measuring insoles (Vertical GRFs), (2) optical MOCAP (3D position of markers), and (3) wearable IMUs (3D linear acceleration/angular velocity/magnetic field; 4D orientation). All three raw datasets were pre-processed in MATLAB. Datasets #1 and #2 (Insoles and MOCAP) were inputted to the OpenSim workflow (I-S; **top**, black). Datasets #1 and #3 (Insoles and IMUs) were inputted to the MATLAB wearable-ANN workflow (W-A; **bottom**, grey). Sagittal and frontal hip joint kinematics (angles) and kinetics (moments) were computed using both methods and then were plotted and compared.

**Figure 2 bioengineering-10-00784-f002:**
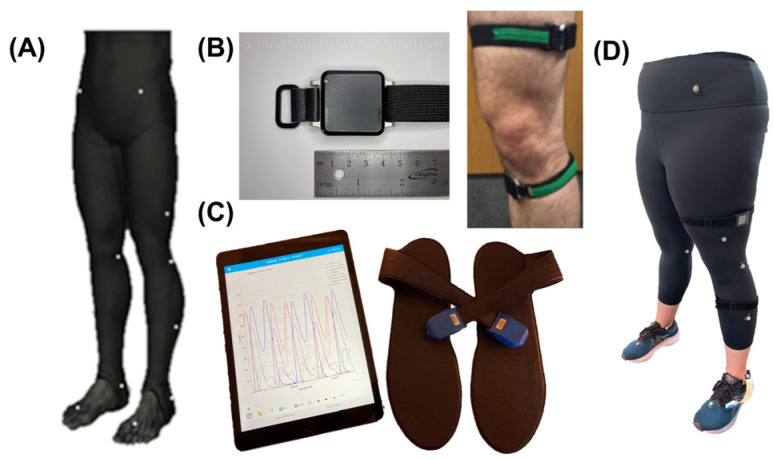
Sensing modalities: (**A**) MOCAP markers (Helen Hayes lower body; 19 markers). (**B**) **Left**: APDM Opal IMU, **Right**: IMU attachment to thigh and shank. (**C**) Loadsol iPad application and force-instrumented insoles. (**D**) Example instrumented subject with all sensing modalities placed appropriately.

**Figure 3 bioengineering-10-00784-f003:**
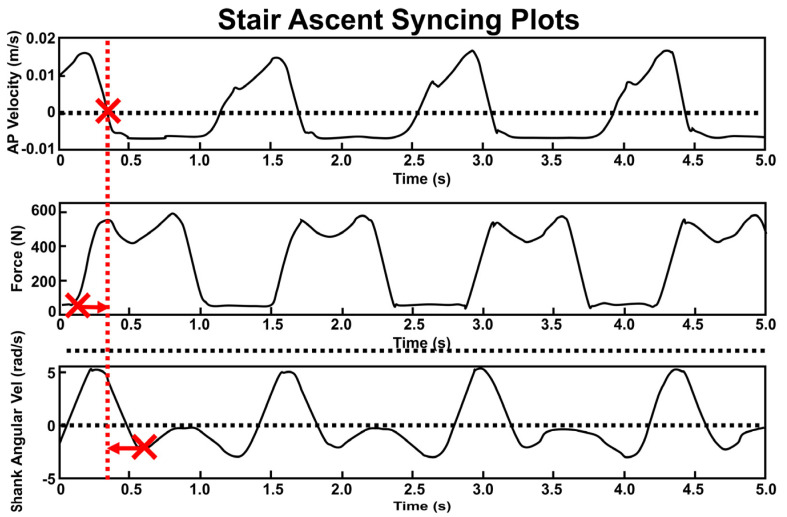
Synchronization of 3 sensing modalities based on first HS time (T_HS_: large Xs). **Top**: anterior−posterior velocity of MOCAP heel marker with dotted line at velocity = 0 m/s. **Middle**: vertical GRF with dotted line at force = 20 N. **Bottom**: IMU angular velocity about the mediolateral axis of shank. T_HS_ from the IMU and insole were shifted to align with MOCAP T_HS_ (indicated by arrows). Vertical red dashed line shows the updated common T_HS_ location after temporal shifting.

**Figure 4 bioengineering-10-00784-f004:**
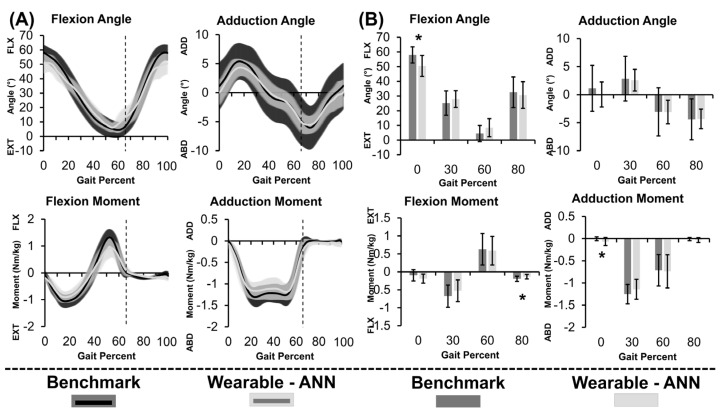
(**A**) Continuous metric data as a percent of the gait cycle and (**B**) Discrete metric data at specific points in the gait cycle displayed as group average hip joint angles (**top**) and moments (**bottom**) computed using gold−standard method (black) and wearable−ANN method (grey) for stair ascent. Toe off at 66% gait cycle is indicated by vertical lines. * *p* < 0.05.

**Figure 5 bioengineering-10-00784-f005:**
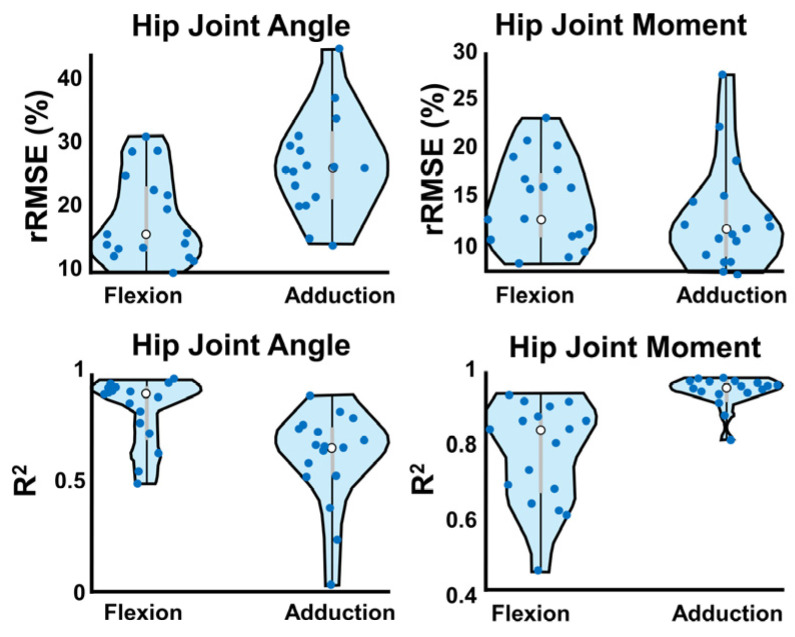
W-A method violin plots of each performance metric. Dots represent the average output for each subject acting as validation across 10 ANN training rounds. White dots are the median, and interquartile range is represented by grey bars. Violin width is determined by fitting a kernel density estimation to show the probability distribution curve.

**Figure 6 bioengineering-10-00784-f006:**
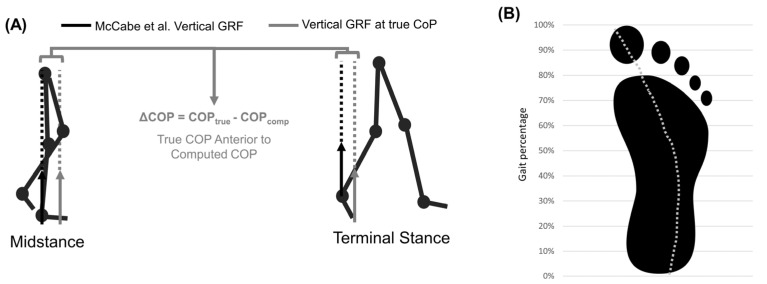
(**A**) Error caused by vertical GRFs being applied to the talus. Solid arrows represent vertical GRF (black = benchmark this study, grey = true CoP vertical GRF). Dotted lines show line of action and lever arm of vertical GRF about hip joint center. (**B**) Center of pressure path during stance.

**Table 1 bioengineering-10-00784-t001:** Descriptive statistics and performance metrics (R^2^ and rRMSE) for sagittal (S) and frontal (F) angle (θ) and moment (µ) using both the benchmark method (B-Mk) and wearable-ANN method (W-ANN). Statistically significant differences (*p* < 0.05) between the B-Mk and W-ANN methods are displayed as bold, italicized text.

Gait (%)	Method	S θ (°)	S µ (Nm/kg)	F θ (°)	F µ (Nm/kg)
0	B-Mk	** *58.0 ± 5.5* **	−0.1 ± 0.2	1.1 ± 4.1	** *−0.003 ± 0.04* **
	W-ANN	** *50.5 ± 7.2* **	−0.2 ± 0.1	0.02 ± 2.2	** *−0.07 ± 0.09* **
30	B-Mk	25.2 ± 8.3	−0.7 ± 0.3	2.8 ± 4.0	−1.3 ± 0.2
	W-ANN	27.9 ± 5.7	−0.5 ± 0.3	2.6 ± 1.9	−1.1 ± 0.2
60	B-Mk	4.5 ± 5.5	0.6 ± 0.4	−3.1 ± 4.3	−0.7 ± 0.4
	W-ANN	8.4 ± 6.2	0.6 ± 0.4	−3.1 ± 2.1	−0.7 ± 0.4
80	B-Mk	32.6 ± 10.4	** *−0.2 ± 0.1* **	−4.4 ± 3.7	−0.01 ± 0.04
	W-ANN	30.7 ± 9.1	** *−0.1 ± 0.1* **	−4.3 ± 1.7	−0.03 ± 0.05
R^2^		0.80 ± 0.02	0.59 ± 0.02	0.77 ± 0.02	0.93 ± 0.01
rRMSE		17.7 ± 1.2	26.4 ± 1.4	14.1 ± 0.8	12.7 ± 1.1

## Data Availability

Due to the large size of the data files in this study, access will be granted on an ad hoc basis.

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
