# Peer review of "Hip Joint Angles and Moments during Stair Ascent Using Neural Networks and Wearable Sensors"

_bioengineering, 2023, doi:10.3390/bioengineering10070784_

Round 1
Reviewer 1 Report
General Comments
This is an interesting manuscript regarding the use of wearable sensors coupled with force measuring insoles and machine learning in the definition of hip kinematics and kinetics. The authors performed a validation study against the gold standard mocap + musculoskeletal modelling assessing the results of a stair ascendant task. The same group reported previous good results of the wearable+ANN approach in the analysis of gait biomechanics.
I believe this is an interesting and well conducted study. The topic of adopting a reduced-cost and -encumbrance setup while maintaining an acceptable level of accuracy in the output data, leveraging ML approaches if needed, is definitely of interest to the readers. Although being a not hidden follow-up study of previous analyses on gait from the same study group, the rationale behind the analysis of a non-level ground activity of daily living is valuable to be assessed on his own.
Methodology description is accurate, although some further details could be added (see specific comments)
Results are properly presented. Figures are helpful to the general understanding, though I suggest increasing the font an image quality in figure 4 (gait percentage intervals could be at 20% stamp instead of 10).
A table summarizing the descriptive and performance results for each parameter should be added, future studies would benefit from a greater level of detail.
I encourage the authors to clarify some key aspects of their paper and correct minor details in order to further improve the quality of their analysis.
Specific Comments
Abstract
- Line 19-20: please add performance results for different variables, do not only report the summary.
- Line 20-21: cannot state "patient-centred” approach based on the findings of the present study. You investigated healthy subjects only, as you correctly pointed out in the rest of the manuscript. Please amend.
Introduction
- L 26: please shorten the introduction. Please also clearly state if the rRMSE <13° that you hypothesized is to be considered for all the variables under investigation.
Methods
- L 142-145: please add detail on the reasons behind this choice.
- L 151-152: please specify that you’re referring to the heel marker here.
- L185: where does this number come from? if 30s*128Hz, it should be 3840. Please clarify
- L 186: please clarify, also in brief, how you output kinematics and kinetics from you 13 IMU inputs.
- L 204-205: please note that rRMSE abbreviation has already been declared previously in the paperò
- L2009-212: I agree with the choice of investigating the performance of the approach at specific time points. However, why not using also a continuous data assessment (e.g., through SPM1D)?
Results
- Please consider adding a table with descriptive and performance outcomes for each variable (see general comments)
Discussion
- Discussion section can be extended. Please recall your rationale and state your main findings first, declaring whether your hypotheses were confirmed or not. The clearly explain the practical implications of your findings before going into the detail of each output and the comparison with previous literature. A conclusion statement is not mandatory but highly recommended.
Reviewer 2 Report
- Line 154: why are you sure that THS correspond to that specific treshold? why exactly 20 N and not related to subject weight? if these are standardized assumptions already proven by previous research, it should be explicitated
- Do the structure of the Stair Master StepMill could have interfered with marker recognition by OptyTrack cameras? this could be a relevant difference compared to analysis performed during standard walk
- Figure 2: If possible, adding a picture of a subject performing the task and wearing all the devices could be much more clear Also, subject were wearing IMUs also on the lower leg. However authors do not mention anything about knee angle and moments, did you collect any data about that?
- Line 235 Maybe Authors could make some hypotesis about this differencies an add it on Discussion section
